# Enhancing bone regeneration and osseointegration using rhPTH(1-34) and dimeric R25CPTH(1-34) in an osteoporotic beagle model

Jeong-Oh Shin[1†], Jong-Bin Lee[2†], Sihoon Lee[3*], Jin-Woo Kim[4*]

[1]Department of Anatomy, Soonchunhyang University College of Medicine, Cheonan, Republic of Korea; [2]Department of Periodontology and Research Institute of Oral Sciences, College of Dentistry, Gangneung-Wonju National University, Gangneung, Republic of Korea; [3]Department of Internal Medicine and Laboratory of Genomics and Translational Medicine, Gachon University College of Medicine, Incheon, Republic of Korea; [4]Department of Oral and Maxillofacial Surgery, Research Institute for Intractable Osteonecrosis of the Jaw, College of Medicine, Ewha Womans University, Seoul, Republic of Korea

*For correspondence:
shleemd@gachon.ac.kr (SL);
jinu600@gmail.com; jwkim84@
ewha.ac.kr (J-WK)

†These authors contributed equally to this work

## eLife Assessment

Using a large animal model, this study demonstrated **valuable** findings that R25CPTH(1-34), based on a mutation associated with isolated familial hypoparathyroidism, generated an anabolic osteointegration effect comparable to that of native PTH1-34. The translational aspect of this human-to-animal work, aimed at animal-to-human translation for therapeutic purposes, should be highlighted. The study design is simple and straightforward, and the methods used are **solid**. The authors have addressed all the questions in their revision.

**Abstract** This study investigates the effects of two parathyroid hormone (PTH) analogs, rhPTH(1-34) and dimeric R25CPTH(1-34), on bone regeneration and osseointegration in a post-menopausal osteoporosis model using beagle dogs. Twelve osteoporotic female beagles were subjected to implant surgeries and assigned to one of three groups: control, rhPTH(1-34), or dimeric R25CPTH(1-34). Bone regeneration and osseointegration were evaluated after 10 weeks using micro-computed tomographic (micro-CT), histological analyses, and serum biochemical assays. Results showed that the rhPTH(1-34) group demonstrated superior improvements in bone mineral density, trabecular architecture, and osseointegration compared to controls, while the dimeric R25CPTH(1-34) group exhibited similar, though slightly less pronounced, anabolic effects. Histological and TRAP assays indicated both PTH analogs significantly enhanced bone regeneration, especially in artificially created bone defects. The findings suggest that both rhPTH(1-34) and dimeric R25CPTH(1-34) hold potential as therapeutic agents for promoting bone regeneration and improving osseointegration around implants in osteoporotic conditions, with implications for their use in bone-related pathologies and reconstructive surgeries.

## Introduction

Bone regeneration is a complex physiological process that is essential for the successful integration of dental implants into the jawbone. Osseointegration, the direct structural and functional connection between living bone and an implant's surface, is a critical determinant of implant stability and long-term viability in dental reconstructive therapy. Since excessive levels of circulating parathyroid hormone (PTH) increase osteoclastic activity and accelerate bone resorption, it might seem paradoxical that PTH can also be used as a treatment modality for diseases with bone loss, such as osteoporosis (*Jilka, 2007*). Administration of PTH analogs can be categorized into two distinct protocols: intermittent and continuous. Intermittent rhPTH(1-34) therapy, typically characterized by daily injections, is clinically used to enhance bone formation and strength. This method leverages the anabolic effects of rhPTH(1-34) without significant bone resorption, which can occur with more frequent or continuous exposure. On the other hand, continuous rhPTH(1-34) exposure, often modeled in research as constant infusion, tends to accelerate bone resorption activities, potentially leading to bone loss (*Silva and Bilezikian, 2015*; *Jilka, 2007*). Understanding these differences is crucial for interpreting the therapeutic implications of rhPTH(1-34) in bone health. Intermittent administration of PTH unlikely continuous exposure showed anabolic effects, indicating different responses relating to bone microarchitecture depending on the dose and frequency (*Silva and Bilezikian, 2015*). However, the underlying mechanisms remain largely unknown, although it was shown that PTH binds through PTH type 1 receptor (PTH1R) and that G-protein-coupled receptors are associated with the protein kinase A-dependent pathway, thereby demonstrating primary anabolic action on bone (*Cheloha et al., 2015*; *Jilka, 2007*). The anabolic effect of intermittent PTH administration is mediated by the downregulation of the Wnt/beta-catenin signaling pathway, which upregulates the transcriptional expression of growth factors, such as IGF1, FGF2, and Runx2, which are essential for the proliferation and differentiation of osteoblasts, leading to an increased number of osteoblasts and survival (*Krishnan et al., 2006*; *Lee and Partridge, 2009*).

A PTH analog, teriparatide (recombinant human PTH(1-34)), demonstrated its promising anabolic effects in a fracture prevention trial (*Neer et al., 2001*), leading to its approval by the United States Food and Drugs Administration, making it the first anabolic agent for postmenopausal osteoporotic women. Additionally, its unique anabolic features, which are contrasted with antiresorptive, lead to an increased application of rhPTH(1-34), whereby it was used in both metabolic and pathological bone diseases alongside various other conditions where bone formation occurred (*Uusi-Rasi et al., 2005*). Conversely, recent concerns regarding the development of osteonecrosis of the jaw (ONJ) have appeared in association with the use of antiresorptive, meaning rhPTH(1-34) has gained attention for its potential to reduce the risk of ONJ and its therapeutic effects in treating ONJ (*Jung et al., 2017*; *Kakehashi et al., 2015*). Although in vivo studies on fracture healing, bone augmentation, and titanium osseointegration effects of rhPTH(1-34) have been attempted, they have generally been limited to pilot studies in rodents (*Gomes-Ferreira et al., 2020*; *Jung et al., 2021*; *Yu and Su, 2020*).

Interestingly, only nine mutations have been discovered since the PTH amino acid and nucleotide sequences were confirmed (*Lee et al., 2020*; *Schipani, 1999*). Among them, the PTH R25C mutation was discovered in three siblings with familial idiopathic hypoparathyroidism, which consisted of a homozygous arginine to cysteine mutation at residue 25 (R25C) in the mature PTH(1-84) polypeptide and exhibited distinct characteristics from the others (*Lee and Lee, 2022*). The other mutations located in the prepro-leader region of the hormone resulted in defective synthesis and secretion; however, the PTH R25C mutation is located within the mature bioactive domain of PTH and does not affect synthesis or secretion (*Lee and Lee, 2022*). Although the capacity of [R25C]PTH(1-34) to bind to the PTH1R and stimulate cAMP production was slightly lower in the human osteoblast-derived SaOS-2 cell line (*Lee and Lee, 2022*), yet it showed comparable anabolic activity in a mouse model (*Bae et al., 2016*). Furthermore, the dimeric formation of the [R25C]PTH(1-34) peptide, presumably through disulfide bonding of the cysteine residues, has implied that dimeric [R25C]PTH(1-34) might partake in unique biological actions, which potentiate its clinical applications (*Park et al., 2021*).

In addition to the limited effectiveness toward certain types of fractures, such as non-vertebral fractures, and the disadvantage of its limited duration of use, the administration of rhPTH(1-34) for osteoporosis has raised concerns about its potential to induce cortical porosity, despite showing favorable results in the treatment of trabecular microarchitecture (*Lindsay et al., 2016*). This has led to concerns regarding the widespread use of PTH. However, recent notable studies have indicated that cortical

porosity is not induced when PTH is administered weekly (*Mosekilde et al., 1995*), which is in contrast to daily administrations (*Yamamoto et al., 2016*; *Zebaze et al., 2017*). These studies suggest that the frequency and dosage of PTH administration can significantly affect the bone response (*Hock and Gera, 1992*), while different forms of PTH might also induce different biological responses, which means deeper investigations into the therapeutic effects of PTH are required (*Bellido et al., 2005*).

Therefore, in this study, the authors used a large animal model that mimics postmenopausal osteoporosis to investigate the therapeutic effects of two PTH analogs, rhPTH(1-34) and dimeric R25CPTH(1-34), on bone regeneration and osseointegration.

## Results

### Microarchitectural and histological analysis of titanium osseointegration

"Our study aims to evaluate and compare the efficacy of rhPTH(1-34) and dimeric R25CPTH(1-34) in promoting bone regeneration and healing using a clinically relevant animal model. Beagle dogs were chosen for their anatomical similarity to human oral structures, suitable size for surgical procedures, human-like bone turnover rates, and well-established oral health profiles, providing a reliable and ethically sound basis for research."

The normal saline-injected control group, the group injected with 40 µg/day PTH (Forsteo, Eli Lilly), and the group injected with 40 µg/day dimeric R25CPTH(1-34) all received subcutaneous injections for 10 weeks.

Various characteristics of the right mandible were evaluated in this analysis, including bone mineral density (BMD), bone volume (BV), trabecular number (Tb.N), trabecular thickness (Tb.Th), and trabecular separation (Tb.Sp) (*Figure 1*). The group administered rhPTH(1-34) presented consistently higher values of BMD, BV, Tb.N, and Tb.Th, and lower value of Tb.Sp compared to the control and dimeric R25CPTH(1-34) groups, indicating that rhPTH(1-34) administration enhanced titanium osseointegration (*Figure 1E*). Moreover, this was consistent for all three titanium implants, irrespective of artificial bone defects (second implant) or bone grafting (third implant). Interestingly, the dimeric R25CPTH(1-34) group showed similar trends for the anabolic effects related to the titanium implants. Morphometric analysis indicated that dimeric R25CPTH(1-34) administration enhanced the BMD, BV, and other trabecular indices, resulting in a higher degree of osseointegration than in the control group (p<0.05), although lower than in the rhPTH(1-34) group (p>0.05) (*Figure 1E*).

Histological analysis further clarified the micro-CT results. The implants in the control group showed physiological bone osseointegration around the titanium implant; however, insufficient bone–implant contact and exposure of implant threads were observed (*Figure 2*, *Figure 2—figure supplement 1*). This was especially evident on the buccal side of the implant, which was more vulnerable due to the bundle bone structure. Alternatively, the titanium implants in the rhPTH(1-34) group were in full contact with the green-stained mineralized bone. The dimeric R25CPTH(1-34) group showed a better pattern of osseointegration compared to the control group, although the bone–implant contact was lower than in the rhPTH(1-34) group (*Figure 2A–C*). Notably, both the rhPTH(1-34) and dimeric R25CPTH(1-34) groups presented evidence of bone regeneration for the second implant, whereby a bone defect was created prior to placing the implant. The measured bone–implant contact ratio was 18.32 ± 16.19% for the control group, 48.13 ± 29.81% for the rhPTH(1-34) group, and 39.53 ± 26.17% for the dimeric R25CPTH(1-34) group, illustrating the significant improvement in osseointegration. (p<0.05 for the control group compared to both PTH groups; however, the difference between the PTH groups was not significant.)

### Histological and TRAP analyses of bone regeneration

Artificial bilateral bone defects were created in the left mandible and the effects of the two PTH analogs were evaluated on the bone regeneration, with one left unfilled and the other filled with a bone graft. *Figure 3* demonstrates the effects of rhPTH(1-34) and dimeric R25CPTH(1-34) on bone regeneration compared to the control group (*Figure 3A–L*). Following the formation of the bone defects, the rhPTH(1-34) and dimeric R25CPTH(1-34) groups achieved sufficient morphological bone regeneration over a period of 10 weeks, while the control group exhibited morphological incompleteness over the same period. The rhPTH(1-34) group exhibited a mature trabecular architecture, while the dimeric R25CPTH(1-34) group showed a similar morphology, although some immature bone formation

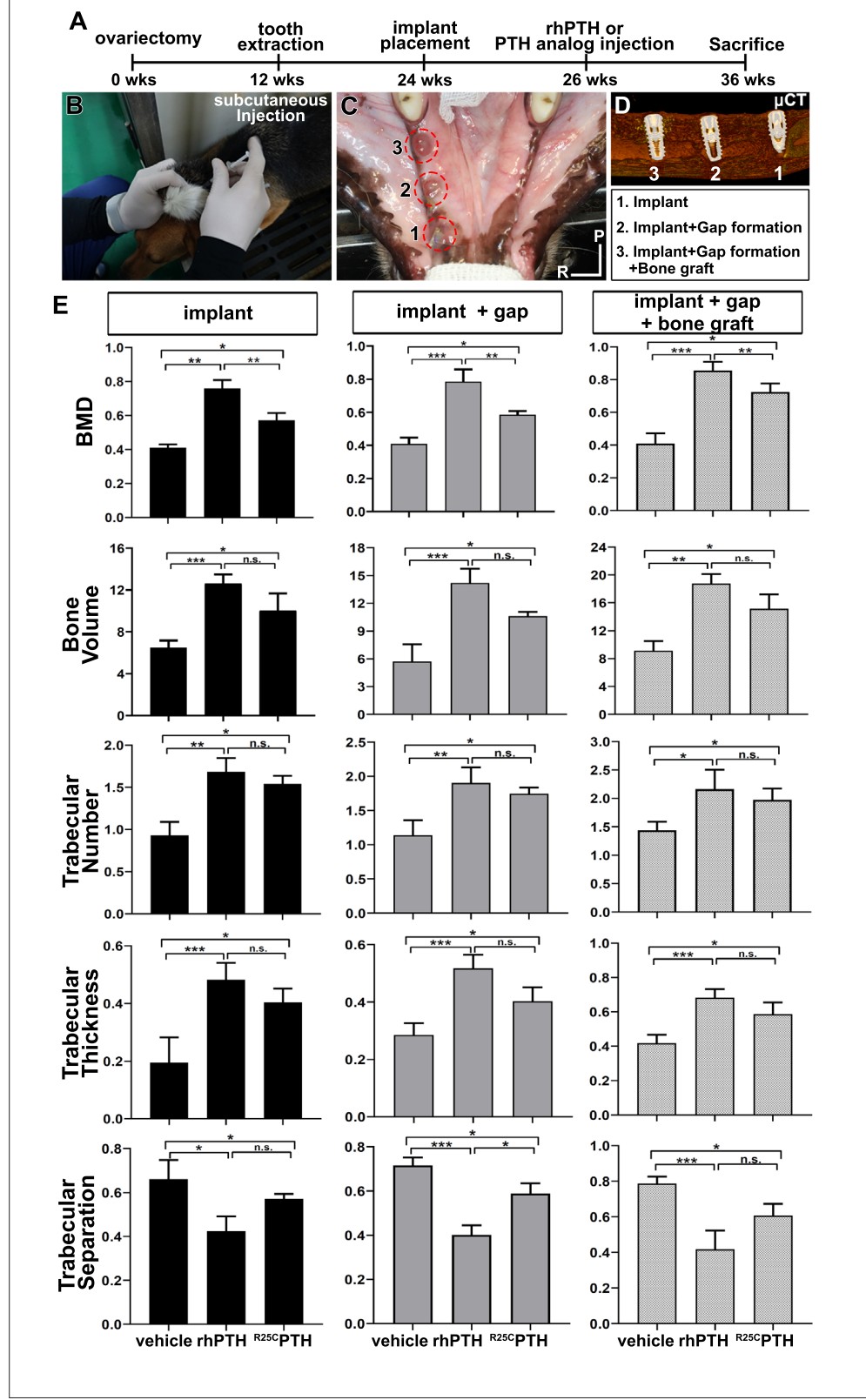

**Figure 1.** Overall Study Design and MicroCT Analysis. (**A–D**) Experimental design for the controlled delivery of rhPTH(1-34) and dimeric $^{R25C}$PTH(1-34) in ovariectomized beagle model. Representative images for injection and place ment of titanium implant. (**E**) Micro-CT analysis. Bone mineral density (BMD), bone volume (TV, mm³), trabecular number (Tb.N, 1 /mm), trabecular thickness (Tb.Th, µm), trabecular separation (Tb.sp, µm). Error bars

*Figure 1 continued on next page*

*Figure 1 continued*

indicate standard deviation. Data are shown as mean ± SD. *p<0.05, **p<0.01, ***p<0.001, n.s., not significant. P, posterior. R, right.

remained stained blue in the Masson trichrome staining analysis (*Figure 3A–L*). While there was no clear difference in the bone defect between the sites with and without bone grafting (*Figure 3E, F, K, and L*), the site where bone grafting occurred exhibited a more mature bone morphology, indicating that a xenograft-maintained space for new bone formation with osteoconductive effects.

The capability of rhPTH(1-34) and dimeric [R25C]PTH(1-34) in bone remodeling were evaluated by tartrate-resistant acid phosphatase (TRAP) immunohistochemical staining (*Figure 4*). Both the rhPTH(1-34) and dimeric [R25C]PTH(1-34) groups showed a significantly higher number of TRAP+ cells at both bone defects, with and without a xenograft, compared to the control group (*Figure 4M and N*) (p<0.05). In addition, the number of TRAP+ cells in the dimeric [R25C]PTH(1-34) group was significantly higher than in the vehicle, yet lower than in the rhPTH(1-34) group (*Figure 4M and N*).

## Serum biochemical analysis

The levels of calcium, phosphorus, CTX, and P1NP were analyzed over time using RM-ANOVA (*Figure 5*). There were no significant differences between the groups for calcium and phosphorus at time points T0 and T1 (*Figure 5A*). However, after the PTH analog was administered at T2 (*Figure 5A*), the levels were highest in the rhPTH(1-34) group, followed by the dimeric [R25C]PTH(1-34) group, and then, lowest in the control group, which was statistically significant (*Figure 5B and C*) (p<0.05). The differences between the groups over time for CTX and P1NP were not statistically significant (*Figure 5D and E*).

## Discussion

This study investigated the therapeutic effects of rhPTH(1-34) and dimeric [R25C]PTH(1-34) on bone regeneration and osseointegration in a large animal model with postmenopausal osteoporosis. rhPTH(1-34) and dimeric [R25C]PTH(1-34) have shown significant clinical efficacy. Although there have been a few studies investigating their effects on bone regeneration in rodents (*Garcia et al., 2013*), we aimed to investigate these effects using a large animal model. We chose this model because it more accurately mimics osteoporotic humans (*Jee and Yao, 2001*). In the evaluation of titanium osseointegration, the rhPTH(1-34) group consistently exhibited enhanced BMD, BV, and other key parameters, thereby indicating superior titanium osseointegration compared to the control and dimeric [R25C]PTH(1-34) groups (*Figure 1E*). Histological analyses confirmed these results, emphasizing the stronger bone–implant contact observed in the rhPTH(1-34) group. Furthermore, both PTH analogs significantly promoted bone regeneration in artificially created defects, with the rhPTH(1-34) group displaying a more mature trabecular architecture, as evidenced by a notable increase in the TRAP+ cell count during the bone remodeling assessments (*Figure 2M and N*).

Regarding the results on the effect of dimeric [R25C]PTH(1-34) in the ovariectomy (OVX) mouse model (*Noh et al., 2024*), bone formation markers were increased in the dimeric [R25C]PTH(1-34) group compared to the rhPTH(1-34) group. Additionally, bone resorption markers were decreased in the rhPTH(1-34) group compared to the control group. However, no significant differences were observed in the dimeric [R25C]PTH(1-34) group. This suggests that the mechanism of action of the dimeric peptide differs from that of the wildtype peptide. Furthermore, based on unpublished data comparing mRNA expression in bone and kidney tissues between the dimeric [R25C]PTH(1-34) and rhPTH(1-34)-treated groups, we strongly believe that dimeric [R25C]PTH(1-34) exhibits distinct biological activity from rhPTH(1-34). These differences may arise from variations in PTH receptor binding, involvement of different G protein subtypes, or downstream intracellular signaling pathways.

The distinct effects of dimeric [R25C]PTH(1-34) and rhPTH(1-34) on osteoblasts and osteoclasts could indicate that while remodeling-based osteogenesis has a limited clinical use period, the dimeric form might promote sustained bone formation and increased bone density over a longer duration. Given that patients with this mutation, who have been exposed to the mutant dimer throughout their lives, exhibit high bone density, this suggests significant potential for dimeric [R25C]PTH(1-34) as a novel therapeutic option alongside wildtype PTH.

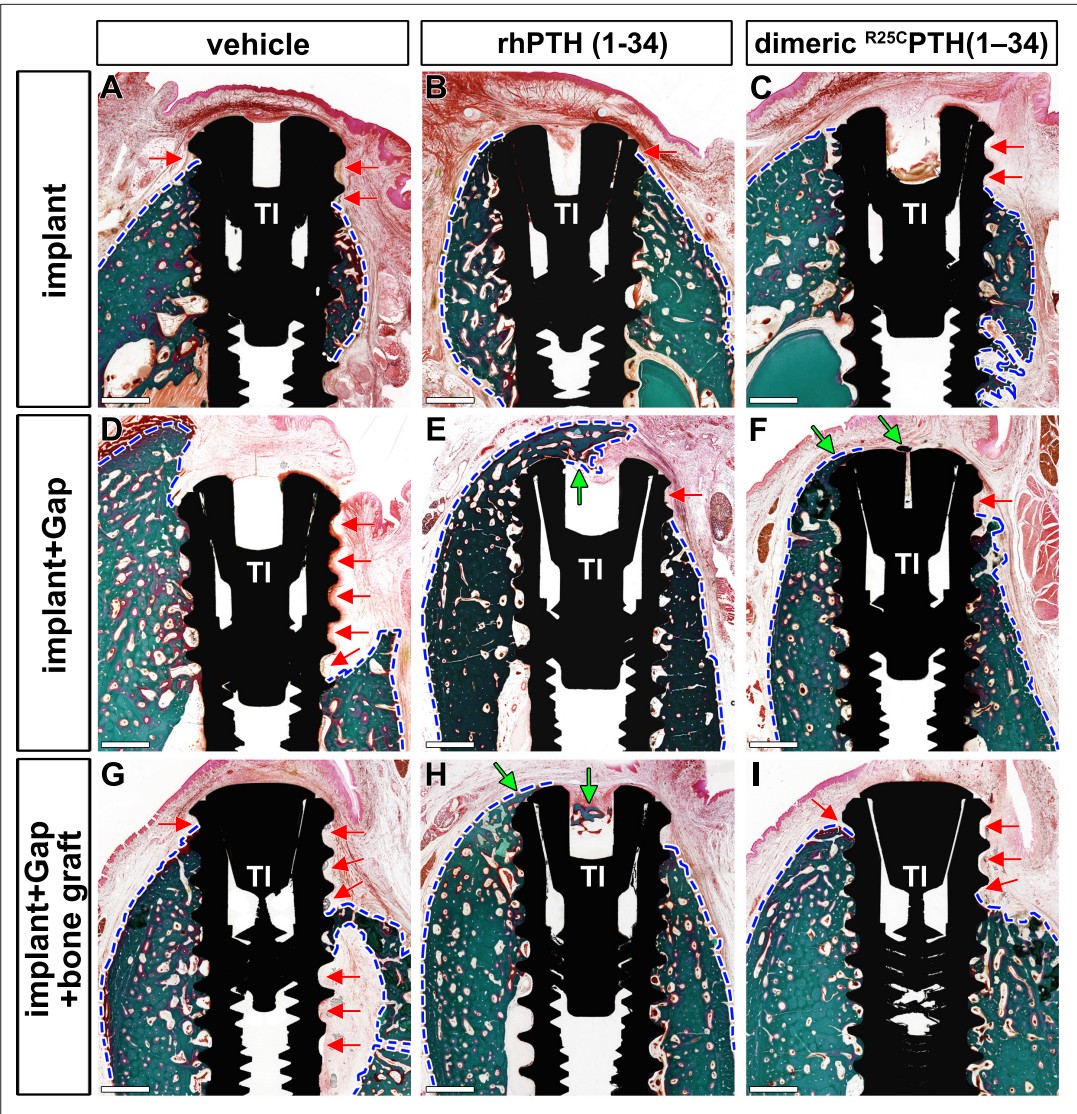

**Figure 2.** Histological Comparison of Bone Formation Across Treatment Groups Using Goldner's Trichrome Staining. (**A–I**) Histological analysis of the different groups stained in Goldner's trichrome. The presence of bone is marked by the green color and soft tissue in red. Red arrows indicate the position with soft tissues without bone around the implant threads. The area of bone formed was the widest in the rhPTH(1-34)-treated group. In the dimeric $^{R25C}$PTH(1-34)-treated group, there is a greater amount of bone than vehicle-treated group. Green arrows represent the bone formed over the implant; blue dotted line, margin of bone and soft tissue; Scale bars, 1 mm.

The online version of this article includes the following figure supplement(s) for figure 2:

**Figure supplement 1.** Three-dimensional reconstructed image of the bone surrounding the implants.

Three-dimensional reconstructed images of the peri-implant bone depicting the osseointegration after different therapeutic interventions. (**A**) Represents the bone response to recombinant human parathyroid hormone fragment (rhPTH 1–34) treatment, showing the most robust degree of bone formation around the implant in the three groups. (**B**) Shows the bone response to a modified PTH fragment (dimeric $^{R25C}$PTH(1-34)), indicating a similar level of bone growth and integration as seen with rhPTH(1-34), although to a slightly lesser extent. (**C**) Serves as the control group, demonstrating the least amount of bone formation and osseointegration. The upper panel provides a top view of the bone–implant interface, while the lower panel offers a cross-sectional view highlighting the extent of bony ingrowth and integration with the implant surface.

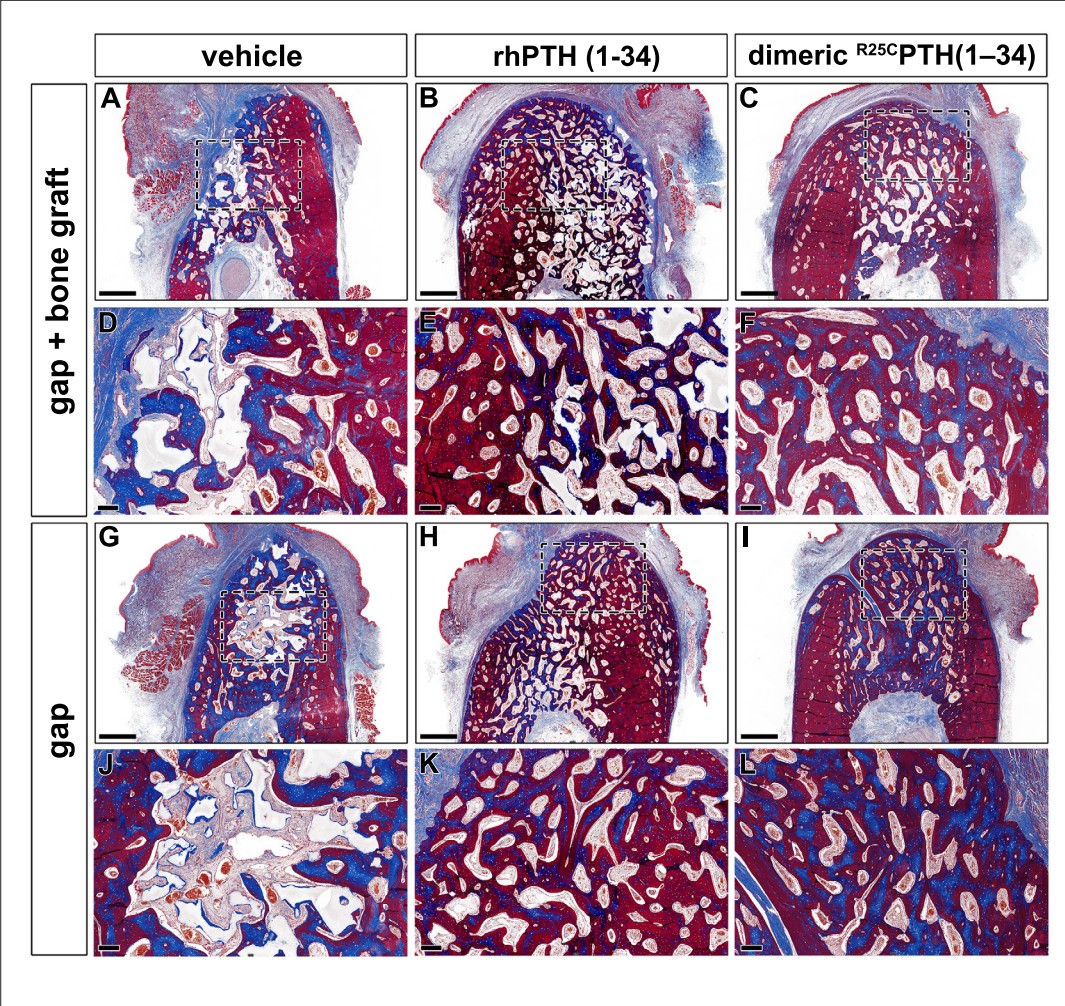

**Figure 3.** Histological analysis using Masson trichrome staining results in the rhPTH(1-34) and dimeric [R25C]PTH(1-34)-treated group. (**A–L**) Masson trichrome-stained sections of cancellous bone in the mandibular bone. The formed bone is marked by the color red. Collagen is stained blue. Black dotted box magnification region of trabecular bone in the mandible. Scale bars, **A–C**, **G–I**: 1 mm; **D–F**, **J–L**: 200 μm.

Both PTH analogs demonstrated anabolic effects, particularly in the mandible. The distinct pharmacological profile of dimeric [R25C]PTH(1-34) suggests potential for more targeted, long-term applications, especially in conditions requiring sustained bone regeneration (*Bae et al., 2016*). This biological difference is thought to be due to dimeric [R25C]PTH(1-34) exhibiting a more preferential binding affinity for the RG versus R0 PTH1R conformation, despite having a diminished affinity for either conformation (*Lee and Lee, 2022*; *Park et al., 2021*). Additionally, the potency of cAMP production in cells was lower for dimeric [R25C]PTH(1-34) compared to monomeric [R25C]PTH(1-34), consistent with its lower PTH1R-binding affinity (*Noh et al., 2024*). One of the potential clinical advantages of dimeric [R25C]PTH(1-34) is its partial agonistic effect in pharmacodynamics. This property may allow for a more fine-tuned regulation of bone metabolism, potentially reducing the risk of adverse effects associated with full agonism, such as hypercalcemia and bone resorption by osteoclast activity. Moreover, the dimeric form may offer a more sustained anabolic response, which could be beneficial in the context of long-term treatment strategies (*Noh et al., 2024*). Also, the impact of dimeric [R25C]PTH(1-34) was notable, as we observed a noticeable improvement in bone formation when compared to the control group. However, these effects were not as strong as those of rhPTH(1-34). Both PTH analogs demonstrated enhanced anabolic effects around the titanium implants, promoting bone regeneration and remodeling.

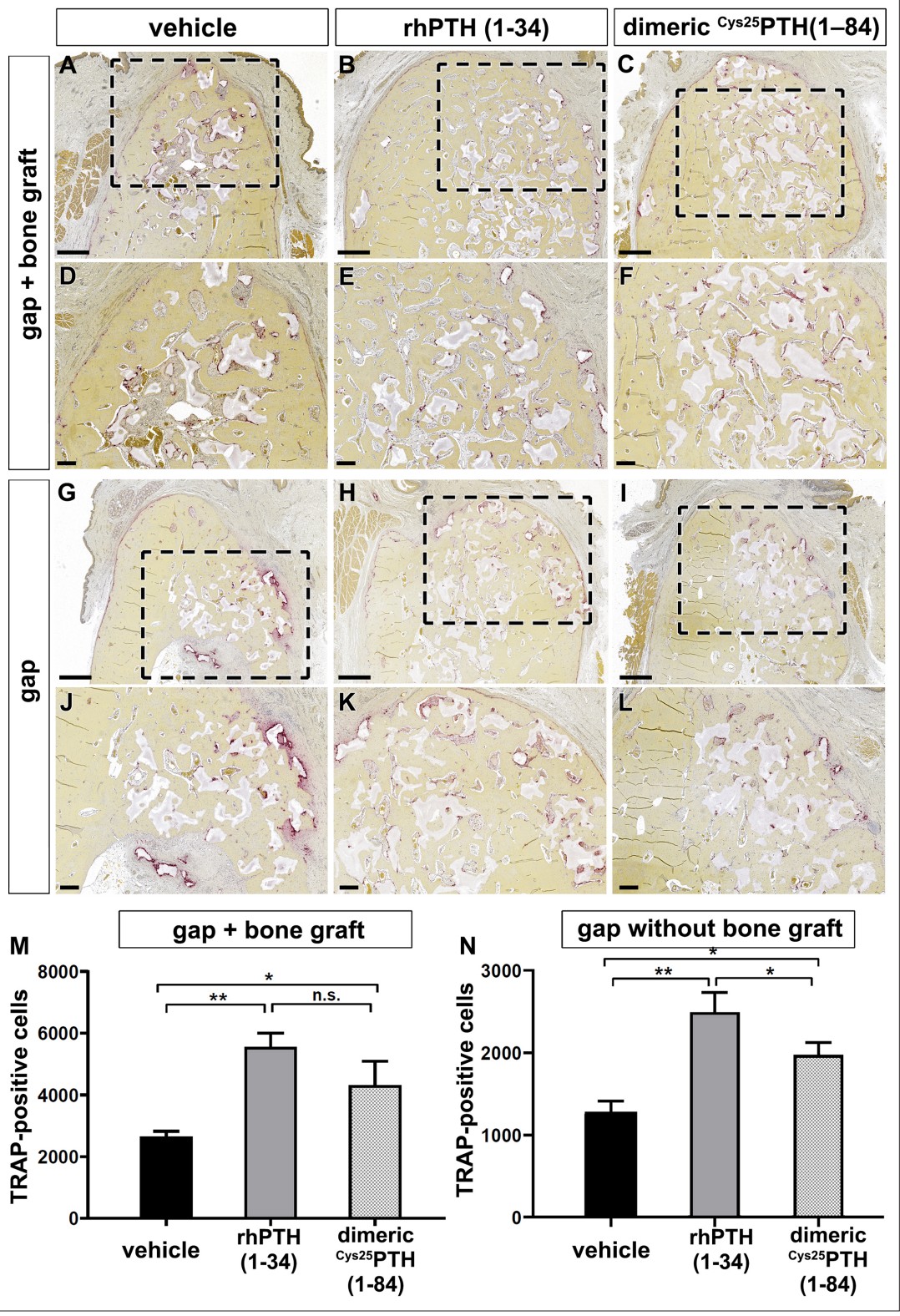

**Figure 4.** Immunohistochemical analysis using tartrate-resistant acid phosphatase (TRAP) staining for bone remodeling activity. (**A–L**) TRAP staining is used to evaluate bone remodeling by staining osteoclasts. Osteoclasts is presented by the purple color. Black dotted box magnification region of trabecular bone in the mandible. (**M, N**) The number of TRAP-positive cells in the mandible with and without xenograft in the rhPTH(1-34) and dimeric [R25C]PTH(1-34)-treated beagle groups. Scale bars, A–C, G–I: 1 mm; D–F, J–L: 200 µm. Error bars indicate standard deviation. Data are shown as mean ± SD. *p<0.05, **p<0.01, n.s., not significant.

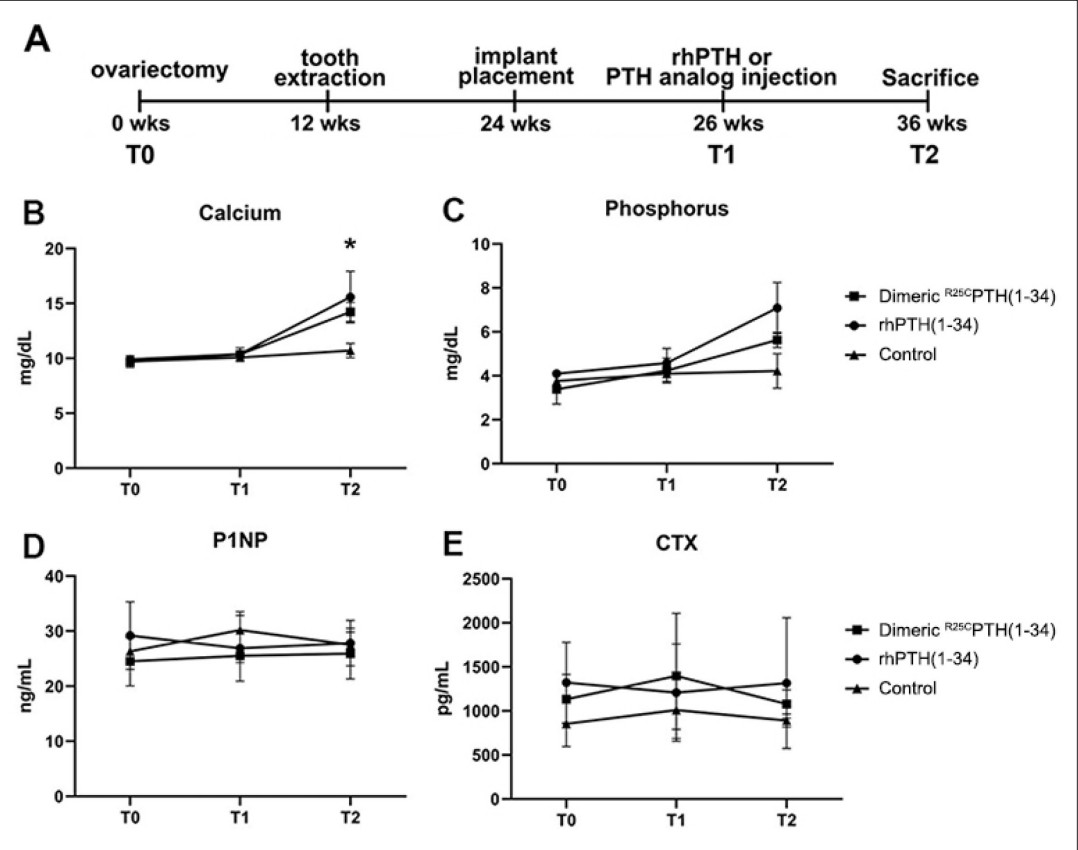

**Figure 5.** Measurement of biochemical Marker Dynamics in serum. The serum levels of calcium, phosphorus, P1NP, and CTX across three time points (**T0, T1, T2**) following treatment with dimeric [R25C]PTH(1-34), rhPTH(1-34), and control. (**A**) The study timeline. (**B–C**) Calcium and phosphorus levels show an upward trend in response to both parathyroid hormone (PTH) treatments compared to control, indicating enhanced bone mineralization. (**D**) P1NP levels, indicative of bone formation, remain relatively stable across time and treatments. (**E**) CTX levels, associated with bone resorption, show no significant differences between groups. Data points for the dimeric [R25C]PTH(1-34), rhPTH(1-34), and control are marked by squares, circles, and triangles, respectively, with error bars representing confidence intervals.

In addition to the effect of systemic bone mineral gain, the unique anabolic feature of PTH has received clinical attention as an emerging strategy due to the increased risk of ONJ following the use of antiresorptive and the rising need for implantation and bone augmentation in the field of orthopedics and maxillofacial surgery (*Ruggiero et al., 2022*). Moreover, site-specific differential effects of teriparatide have not been clarified, although it represents an issue often associated with selective concentrations of teriparatide that cause anabolic effects on the central skeleton and possible bone mineral decrease on the peripheral skeleton, including the skull (*McClung et al., 2005*). Moreover, it can be inferred that facial and jawbones, which have the same developmental origin as the skull through membranous ossification, will show the same bone response as the skull (*Setiawati and Rahardjo, 2019*). Previous studies have demonstrated that the central skeleton, including the lumbar and thoracic spine and pelvis regions, showed an increase in areal BMD, while the arms, legs, and skull showed a decrease in bone minerals, suggesting an effect of rhPTH(1-34) on the redistribution of bone minerals from the peripheral to the central skeleton (*Paggiosi et al., 2018*).

However, the results of this study demonstrated that rhPTH(1-34) and dimeric [R25C]PTH(1-34) significantly improved the osseointegration of bone and titanium, as well as jawbone regeneration (*Figure 3*). The anabolic effects of both PTH analogs in this specific region may have been enhanced by the unique anatomical characteristics of the mandible, which we attribute to these improvements. The authors have attributed this phenomenon to the unique anatomical characteristics observed in the jawbone. The jawbone in the human body undergoes the most rapid bone remodeling and has

excellent blood flow (*Huja et al., 2006*). Since the jawbone is continuously exposed to mechanical stress from mastication and swallowing, this suggests that the net anabolic effect of rhPTH(1-34) in the jawbone, which is not part of the central skeleton, is achieved through mechanical loading. A recent study by *Robinson et al., 2021* demonstrated that rhPTH(1-34) and mechanical loading additively stimulate anabolic modeling and synergistically stimulate remodeling in trabecular bone, findings that further support this notion (*Robinson et al., 2021*). However, further investigation is needed to fully understand this relationship.

The anabolic effects of rhPTH(1-34) have been demonstrated through several large randomized controlled trials (*Neer et al., 2001*; *Tsai et al., 2013*). Despite the FDA's decision to remove the 2-year treatment limit in 2021, which opens possibilities for broader clinical applications, there are still numerous challenges that need to be addressed. There are ongoing concerns about the potential long-term effects of extended use, including accelerated bone remodeling, possible hypercalcemic conditions, and heightened bone resorption (*Burr et al., 2001*; *Fox et al., 2007a*; *Fox et al., 2007b*; *Neer et al., 2001*; *Sato et al., 2004*; *Tsai et al., 2013*). However, recent studies have shown that the frequency and dosing of rhPTH(1-34) administration can lead to different bone responses (*Yamamoto et al., 2016*; *Yamane et al., 2017*). An in vivo study by Yamamoto et al. reported that lower doses of rhPTH(1-34) with high-frequency administration resulted in the formation of thin trabeculae, osteoclastogenesis, and accelerated bone remodeling, while low-frequency rhPTH(1-34) administration showed a phenomenon of modeling-based formation through thicker trabeculae, mature osteoblasts, and new bone formation (*Yamane et al., 2017*). This supports the notion that modeling-based bone gain can be an important axis in PTH anabolism and the primary principle of PTH as bone remodeling-driven bone anabolism.

The use of daily injections in this study was intended to simulate intermittent PTH therapy, a well-established clinical approach for managing osteoporosis and enhancing bone regeneration. Intermittent administration of PTH, as opposed to continuous exposure, is critical for maximizing the anabolic response while minimizing the catabolic effects that are associated with higher frequency or continuous hormone levels. Our findings support the notion that even with daily administration, both rhPTH(1-34) and dimeric [R25C]PTH(1-34) promote bone formation and osseointegration, consistent with the outcomes expected from intermittent therapy. It's important for future research to consider the dosage and timing of administration to further optimize the therapeutic benefits of PTH analogs (*Dempster et al., 2013*; *Hodsman et al., 2005*).

The limitation of this study is that the therapeutic responses of rhPTH(1-34) and dimeric [R25C]PTH(1-34) were focused on local surgical interventions, meaning that we could not investigate the central skeletal responses, such as in the femur and lumbar. Therefore, further research is needed to investigate these different responses by administering the PTH analogs at various frequencies and doses. Additionally, although this study was conducted on large animals, using a larger number of subjects would have yielded more robust results. Implementing randomization of the surgical sites and using guided surgery for implant insertion should also be considered in future studies to achieve more precise results.

Overall, the study demonstrated the therapeutic effects of rhPTH(1-34) and dimeric [R25C]PTH(1-34) on bone regeneration and titanium osseointegration using a beagle model with osteoporosis. Validation of the anabolic effects of rhPTH(1-34) and dimeric [R25C]PTH(1-34) in large animals has resulted in a broader understanding of their physiological and therapeutic functions and further expands their potential applications.

## Materials and methods
### Animal preparation
This study was conducted in compliance with the ARRIVE guidelines and approved by the Animal Research Committee of Cronex Co., Ltd., Hwaseong, South Korea (CRONEX-IACUC 201801002). All animal experiments, including animal selection, management, preparation, and surgical protocols, were conducted in compliance with the Ewha Womans University rules for animal experiments. The animals were housed in a standard laboratory environment (21±1°C with 40–70% humidity; 12 hr light/dark cycle) with a standardized food/water supply.

Beagles were chosen for this study because the bone size and dentition could accommodate human dental implants and the application of mechanical force to implants. To induce osteoporosis, 12 female beagles underwent bilateral OVX at 12 weeks of age, followed by osteoporosis development for 12 weeks before being used for the next experiments at the age of 24 weeks. Health and oral hygiene were checked and maintained daily (*Figure 1A*).

## Experiment protocol

Anesthesia was induced using zolazepam/tiletamine (10 mg/kg body weight, Zoletil; Virbac Laboratories, Carros, France) and xylazine hydrochloride (Rumpun, Bayer, Leverkusen, Germany), by intramuscular injection. The beagles were anesthetized using inhalation anesthesia for the implant surgery, and antibiotics (Ceftriaxone, Kyungdong Pharm, Seoul, South Korea) were administered for 3 days.

Both mandibular premolars 1–4 were extracted at 12 weeks after OVX. Then, 12 weeks later, three dental titanium implants (TS III 3.0×10 mm, Osstem, Seoul, Korea) were inserted into the right lower jaw of each animal via a conventional implant surgical procedure under saline irrigation. Each implant was placed over at least a 3.0 mm distance. In detail, the first implant was placed over 3.0 mm behind the canine. The second and third implants were inserted to create a 3 mm circumferential bony defect using a Ø 6 mm trephine bur. Then, an additional bone graft (Bio-oss small particle 1.0 g and Bio-gide, Geistlich, Switzerland) was placed on the bone defect around the third implant, and the bone gap remained with nothing at the second implant. On the left lower jaw, two artificial bone defects of 5×10 mm were made using trephine bur. The anterior hole remained defective without a graft, while a bone graft (Bio-oss small particle 1.0 g and Bio-gide, Geistlich, Switzerland) was applied to the posterior hole.

After 2 weeks of healing, 12 female OVX beagles were randomly designated into three groups as follows: (1) control group with normal saline injection, (2) PTH (1-34) group with daily 40 μg/day rhPTH(1-34) injection (Forsteo, Eli Lilly and Company, Indianapolis, IN, USA), and (3) dimeric R25CPTH(1-34) group with 40 μg/day injection (dimeric R25CPTH(1-34), chemically synthesized by the Anygen, Gwangju, Republic of Korea). Each animal received one injection per day, aimed at replicating the intermittent rhPTH(1-34) exposure proven beneficial for bone regeneration and overall skeletal health in clinical settings (*Neer et al., 2001*; *Kendler et al., 2018*). This regimen was chosen to investigate the potential anabolic effects of these specific PTH analogs under conditions closely resembling therapeutic use. Animals were injected subcutaneously for 10 weeks, after which, they were euthanized and the bone regeneration and implant osseointegration were evaluated (*Figure 1A–C*).

## Micro-CT analysis

To examine the microarchitectural effects of rhPTH(1-34) and dimeric R25CPTH(1-34) on bone regeneration and osseointegration, radiographic analysis was performed using micro-CT on the right mandible (*Figure 1D*). The specimens were fixed in 4% paraformaldehyde for 48 hr before being assessed by micro-CT (SkyScan1173 ver. 1.6, Bruker-CT, Kontich, Belgium). The specimens were imaged with a pixel size of 29.83 μm. The voltage and current intensities of the images were 130 kV and 60 μA, respectively. The regions of interest were determined as the $10 \times 10$ mm$^2$ square area, located 3 mm from the bottom of the implant. BMD, BV (mm$^3$), Tb.N (1/mm), Tb.Th (μm), and Tb.Sp (μm) were analyzed.

## Histological and histomorphometric analysis

Bone histomorphometric parameters were computed and shown in accordance with recommendations by the ASBMR histomorphometric nomenclature committee (*Dempster et al., 2013*). Goldner's trichrome and Masson trichrome staining were performed on both the right implantation and left bone defect sites, respectively. Specimens were dehydrated in increasing concentrations of ethanol and embedded in a mixture of ethanol and Technovit 7200 resin (Heraeus Kulzer, Wehrheimm, Germany), with an increasing ratio of resin. Following resin infiltration, the specimens were hardened in a UV embedding system (KULZER EXAKT 520, Norderstedt, Germany) for a day. The undecalcified specimens were cut using an EXAKT diamond cutting system (EXAKT 300 CP, Norderstedt, Germany), and the soft tissue and bone were attached to an acryl slide by an adhesive system. The section width of the specimen was adjusted to 40±5 μm using a grinding system (EXAKT 400CS, KULZER, Norderstedt, Germany). The specimens on the right implantation site were stained with Goldner's trichrome

and photographed by a Panoramic 250 Flash III system (3DHISTECH Ltd., Budapest, Hungary). The bone–implant contact ratio (%) was assessed as the linear percentage of the interface with direct contact between the bone and implant to the total interface of the implant using CaseViewer program software (3DHISTECH Ltd.).

TRAP assay was performed on the left bone defect sites. The bone specimens were fixed in 4% paraformaldehyde overnight and decalcified in 10% ethylenediaminetetraacetic acid for 7 days. The decalcifying solution was changed every other day. The specimens were embedded in paraffin and cut into sections. According to the manufacturer's instructions, the sections were deparaffinized and stained using a TRAP staining kit (Sigma, St. Louis, MO, USA). The number of TRAP-positive cells in the sections was counted under a microscope (DM2500, Leica Microsystems, Wetzlar, Germany).

### Serum biochemical analysis

Fasting blood samples were drawn in the morning at baseline (T0; 12 weeks), the start of the injection at 26th week (T1), and euthanasia at 36th week (T2) for P1NP (procollagen type I N-terminal propeptide), PTH, CTX (C-terminal telopeptide), and calcium and phosphorus (*Figure 5A*). Calcium and phosphorus were analyzed using the Beckman AU480 Chemistry Analyzer (Beckman Coulter AU480). P1NP (procollagen type I N-terminal propeptide ELISA kit, Mybiosource), PTH (parathyroid hormone ELISA kit, Aviva Systems Biology), and CTX (C-terminal telopeptide [CTx-I] ELISA kit, Mybiosource) were analyzed by the ELISA method, according to the ELISA kit manufacturer's instructions. In all analyses, the measured values were below the limit of quantification for the standard curve.

### Statistical analysis

Data for microarchitectural, histomorphometric, and serum biochemical analyses were expressed as mean and standard deviation (SD). Non-parametric tests were performed, including the Mann–Whitney and Kruskal–Wallis tests. Group differences in serum markers over time were compared by repeatedly measuring the analysis of variance. Statistical analysis was performed using SPSS 26 (IBM Corp., USA) and Prism 10 (GraphPad, San Diego, CA, USA). p-Values of <0.05 were set as statistically significant.

## Acknowledgements

The authors declare no conflict of interest. This work was supported by a grant from the Korea Health Technology R&D Project through the Korea Health Industry Development Institute (KHIDI), funded by the Ministry of Health & Welfare, Republic of Korea (HI22C1377) and the National Research Foundation of Korea (NRF) grant funded by the Korean government (MSIT) (No. 2022R1A2C3006002, RS-2024-00339519 and 2018R1D1A1B07041400). This work was supported by the Gachon University Gil Medical Center (FRD2023-12), the Gachon University research fund of 2023(GCU-202309040001) and Soonchunhyang University Fund (No. 20220446).

## Additional information

### Competing interests

Sihoon Lee: Reviewing editor, *eLife*. The other authors declare that no competing interests exist.

### Funding

| Funder | Grant reference number | Author |
| --- | --- | --- |
| Korea Health Industry Development Institute | HI22C1377 | Jin-Woo Kim |
| National Research Foundation of Korea | RS-2024-00339519 | Jin-Woo Kim |
| National Research Foundation of Korea | 2022R1A2C3006002 | Sihoon Lee |

| Funder | Grant reference number | Author |
|---|---|---|
| National Research Foundation of Korea | 2018R1D1A1B07041400 | Jong-Bin Lee |
| Gachon University Gil Medical Center | FRD2023-12 | Sihoon Lee |
| Soonchunhyang University Fund | 20220446 | Jeong-Oh Shin |

The funders had no role in study design, data collection and interpretation, or the decision to submit the work for publication.

### Author contributions

Jeong-Oh Shin, Conceptualization, Resources, Formal analysis, Funding acquisition, Validation, Investigation, Methodology, Writing – original draft, Writing – review and editing; Jong-Bin Lee, Resources; Sihoon Lee, Conceptualization, Resources, Supervision, Validation, Writing – original draft, Writing – review and editing; Jin-Woo Kim, Conceptualization, Resources, Data curation, Formal analysis, Supervision, Funding acquisition, Validation, Investigation, Visualization, Methodology, Writing – original draft, Writing – review and editing

### Author ORCIDs

Jong-Bin Lee ⓘ https://orcid.org/0000-0002-6800-4337
Sihoon Lee ⓘ https://orcid.org/0000-0002-9444-5849
Jin-Woo Kim ⓘ https://orcid.org/0000-0002-1672-5730

### Ethics

This study was conducted in compliance with the ARRIVE guidelines and approved by the Animal Research Committee of Cronex Co., Ltd., Hwaseong, South Korea. (CRONEX-IACUC 201801002).

Reviewer #1 (Public Review): https://doi.org/10.7554/eLife.93830.5.sa1
Reviewer #2 (Public Review): https://doi.org/10.7554/eLife.93830.5.sa2
Reviewer #3 (Public Review): https://doi.org/10.7554/eLife.93830.5.sa3
Author response https://doi.org/10.7554/eLife.93830.5.sa4

## Additional files

### Supplementary files
• MDAR checklist

### Data availability

All relevant data has been made publicly available on Dryad at https://doi.org/10.5061/dryad.hmgqnk9t1.

The following dataset was generated:

| Author(s) | Year | Dataset title | Dataset URL | Database and Identifier |
|---|---|---|---|---|
| Kim J-W, Shin J-O, Lee J-B, Lee S | 2024 | Measurement of biochemical Marker Dynamics in serum levels of calcium, phosphorus, P1NP, and CTX across three time points (T0, T1, T2) following treatment with dimeric R25CPTH(1-34), rhPTH(1-34), and control | https://doi.org/10.5061/dryad.hmgqnk9t1 | Dryad Digital Repository, 10.5061/dryad.hmgqnk9t1 |

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
