## [Editor Report · eLife Assessment]

Using a large animal model, this study demonstrated **valuable** findings that ^R25C^PTH(1-34), based on a mutation associated with isolated familial hypoparathyroidism, generated an anabolic osteointegration effect comparable to that of native PTH1-34. The translational aspect of this human-to-animal work, aimed at animal-to-human translation for therapeutic purposes, should be highlighted. The study design is simple and straightforward, and the methods used are **solid**. The authors have addressed all the questions in their revision.

---

## [Referee Report · Reviewer #1 (Public Review)]

Summary:

This study provides valuable insights into the therapeutic effects of two parathyroid hormone (PTH) analogs on bone regeneration and osseointegration. The research is methodologically sound, employing a robust animal model and a comprehensive array of analytical techniques, including micro-CT, histological/histomorphometric analyses, and serum biochemical analysis.

Strengths:

The use of a large animal model, which closely mimics postmenopausal osteoporosis in humans, enhances the study's relevance to clinical applications. The study is well-structured, with clear objectives, detailed methods, and a logical flow from introduction to conclusion. The findings are significant, demonstrating the potential of rhPTH(1-34) and dimeric R25CPTH(1-34) in enhancing bone regeneration, particularly in the context of osteoporosis.

Weaknesses:

There are no major weaknesses.

---

## [Referee Report · Reviewer #2 (Public Review)]

Summary:

This article explores the regenerative effects of recombinant PTH analogues on osteogenesis.

Strengths:

Although PTH has known to induce the activity of osteoclasts, accelerating bone resorption, paradoxically its intermittent use has become a common treat for osteoporosis. Previous studies successfully demonstrated this phenomenon in vivo, but most of them used rodent animal models, inevitably having a limitation. In this article, the authors tried to address this, using a beagle model, and assessed the osseointegrative effect of recombinant PTH analogues. As a result, the authors clearly observed the regenerative effects of PTH analogues, and compared the efficacy, using histologic, biochemical, and radiologic measurement for surgical-endocrinal combined large animal models. The data seem to be solid, and has potential clinical implications.

Weaknesses:

All the issues that I raised have been resolved in the revision process.

Overall, this paper is well-written and has clarity and consistency for a broader readership.

---

## [Referee Report · Reviewer #3 (Public Review)]

Summary:

The work submitted by Dr. Jeong-Oh Shin and co-workers aims to investigate the therapeutic efficacy of rhPTH(1-34) and R25CPTH(1-34) on bone regeneration and osseointegration of titanium implants using a postmenopausal osteoporosis animal model.

In my opinion the findings presented are not strongly supported by the provided data since the methods utilized do not allow to significantly support the primary claims.

Strengths:

Strengths include certain good technologies utilized to perform histological sections (i.e. the EXAKT system).

Weaknesses:

Certain weaknesses continue to significantly lower the enthusiasm for this work. Most important: the limited number of samples/group. In fact, as presented, the work has an n=4 for each treatment group. This limited number of samples/group significantly impairs the statistical power of the study. In addition, the implants were surgically inserted following a "conventional implant surgery", implying that no precise/guided insertion was utilized. This weakness is, in my opinion, particularly significant since the amount of bone osteointegration may greatly depend on the bucco-lingual positioning of each implant at the time of the surgical insertion (which should, therefore, be precisely standardized across all animals and for all surgical procedures).

---

## [Author Response]

The following is the authors’ response to the previous reviews.

**Reviewing Editor (Recommendations For The Authors):**
The revised manuscripts and rebuttal sufficiently answered all the questions raised by three Reviewers. Overall, the manuscript is well written and the results are clear based on a straightforward experiment in a pursuit of comparing dimeric PTH analog to 1-34 PTH analog, which has established clinical efficacy. The study's results are valuable as it utilized large animal models, specifically examined the local bone integration effects, and demonstrated the comparable therapeutic efficacy of the new PTH analog to 1-34 PTH. However, the data did not convincingly show how the dimeric PTH analog overcomes the limitations of 1-34 PTH. I suggest that the discussion should focus more on the differences between the two analogs.

We sincerely appreciate your thorough review and valuable feedback. We have carefully considered your comments and would like to address them as follows:

“Regarding the results on the effect of dimeric R25CPTH(1-34) in the OVX mouse model (Noh et al., 2024), bone formation markers were increased in the dimeric R25CPTH(1-34) group compared to the rhPTH (1-34) group. Additionally, bone resorption markers were decreased in the rhPTH (1-34) group compared to the control group. However, no significant differences were observed in the dimeric R25CPTH(1-34) group. This suggests that the mechanism of action of the dimeric peptide differs from that of the wildtype peptide. Furthermore, based on unpublished data comparing mRNA expression in bone and kidney tissues between the dimeric R25CPTH(1-34) and rhPTH (1-34) treated groups, we strongly believe that dimeric R25CPTH(1-34) exhibits distinct biological activity from rhPTH (1-34). These differences may arise from variations in PTH receptor binding, involvement of different G protein subtypes, or downstream intracellular signaling pathways.

The distinct effects of dimeric R25CPTH(1-34) and rhPTH (1-34) on osteoblasts and osteoclasts could indicate that while remodeling-based osteogenesis has a limited clinical use period, the dimeric form might promote sustained bone formation and increased bone density over a longer duration. Given that patients with this mutation, who have been exposed to the mutant dimer throughout their lives, exhibit high bone density, this suggests significant potential for dimeric R25CPTH(1-34) as a novel therapeutic option alongside wildtype PTH.” (Discussion section 2nd paragraph)

A few minor points I 'd like to point out. This line number is based on a Word file.Line 146-148 - However, both were insufficient compared to the control group and did not illustrate any bone filling. The measured bone-implant contact ratio was 18.32 {plus minus} 16.19% for the control group, 48.13 {plus minus} 29.81% for the group, and 39.53 {plus minus} 26.17% (P < 0.05).- Does it mean that bone generation of both treatment group is inferior to the control group? please specify which groups the values are belong to and between which groups P-value compare.

Thank you very much for your suggestion to improve the manuscript. We have recognized the previous omission and have revised the sentence clearly as follows.

"The measured bone–implant contact ratio was 18.32 ± 16.19% for the control group, 48.13 ± 29.81% for the rhPTH(1-34) group, and 39.53 ± 26.17% for the dimeric R25CPTH(1-34) group, illustrating the significant improvement in osseointegration. (P < 0.05 for the control group compared to both PTH groups; however, the difference between the PTH groups was not significant.)"

Line 157 - incompleteness over the same period. The rhPTH(1-34) group exhibited a mature trabecularcfghnc- Please correct misspellings.

As the reviewer mentioned, I have corrected "trabecularcfghnc" to "trabecular." Thank you.

Line 165-168 and Figure 4 M-N - Both the rhPTH(1-34) and dimeric R25CPTH(1-34) groups showed a significantly higher number of TRAP+ cells at both bone defects, with and without a xenograft, compared to the control group (Figure 4M,N). (P < 0.05) In addition, the number of TRAP+ cells in the dimeric R25CPTH(1-34)group was significantly higher than in the vehicle, yet lower than in the rhPTH(1-34) group (Figure 4M,N).- I believe the heading of figure 4M-N should be changed to with or without xenograft. And maybe you want to explain the significant difference of TRAP positive cells between two groups (with vs. without xenograft). Minor point: was - were

We totally agree with reviewer’s comment. We changed figure 4. Also, based on the revised figure, the figure legends for figure 4 were also revised as follows. “The number of TRAP-positive cells in the mandible with and without xenograft in the rhPTH(1-34) and dimeric R25CPTH(1-34)-treated beagle groups.” Following the reviewer's comments, the be verb in the sentences in the results section was changed from ‘was’ to ‘were’. “The capability of rhPTH(1-34) and dimeric R25CPTH(1-34) in bone remodeling were evaluated by tartrate-resistant acid phosphatase (TRAP) immunohistochemical staining.”

Line 182-186 - This study investigated the therapeutic effects of rhPTH(1-34) and dimeric R25CPTH(1-34) on bone regeneration and osseointegration in a large animal model with postmenopausal osteoporosis. rhPTH(1-34) and dimeric R25CPTH(1-34) have shown significant clinical efficacy, and although there have been a few studies investigating their effects on bone regeneration in rodents (Garcia et al., 2013), the authors in this study aimed to investigate the effects using a large animal model that more accurately mimics osteoporotic humans (Cortet, 2011).- Please split the sentences for better clarity. In last sentence, I'm unsure what Cortet 2011 citation here is for. The statement should be written in the first person not the third person.

We appreciate your attention to detail, which has helped improve the clarity and accuracy of this manuscript. As per the reviewer's suggestion, I have reordered and changed the references to fit the content and revised the sentences to the first person.

“rhPTH(1-34) and dimeric R25CPTH(1-34) have shown significant clinical efficacy. Although there have been a few studies investigating their effects on bone regeneration in rodents (Garcia et al., 2013), we aimed to investigate these effects using a large animal model. We chose this model because it more accurately mimics osteoporotic humans (Jee and Yao, 2001).”

Line 196-197 - Furthermore, by demonstrating that dimeric R25CPTH(1-34) exhibits a distinct pharmacological profile different from rhPTH(1-34) but still provides a clear anabolic effect in the localized jaw region, the authors have shown that it may possess different potential therapeutic indications from rhPTH(1-34).- This study does not include any pharmacological data. (Please cite reference). Again, I would suggest writing it in the first person. It sounds like you are reviewing someone else's work

Thank you for your insightful comments. We acknowledge that our study did not include pharmacological data. We have changed the sentence to clarify that the pharmacological profile information is derived from previous studies. A suitable citation was included to substantiate this assertion. As suggested, we have revised the statement in the first person to more accurately represent our own research and discoveries.

“Furthermore, we have shown that dimeric R25CPTH(1-34) has a distinct anabolic effect in the localized mandible region, which is comparable to that of rhPTH(1-34). Our findings indicate that dimeric R25CPTH(1-34) may have distinct potential therapeutic indications, as demonstrated by prior pharmacological studies (Bae et al., 2016), which demonstrated that it possesses a distinct pharmacological profile from rhPTH(1-34).”

Line 201 - One of the potential clinical advantages of dimeric R25CPTH(1-34) is its partial agonistic effect in pharmacodynamics.- it needs reference

Thank you for your insightful advice. As reviewer’s suggestion, we have included references as follows.

“Additionally, the potency of cAMP production in cells was lower for dimeric R25CPTH compared to monomeric R25CPTH, consistent with its lower PTH1R-binding affinity (Noh et al., 2024).”

Line 206-207 - Also, the effects of dimer were prominent, as we mentioned better bone formation than the control groupBut not compared with monomeric 1-34 PTH

We have revised the statement to more accurately reflect our findings.

“Also, the impact of dimeric R25CPTH(1-34) was notable, as we observed a noticeable improvement in bone formation when compared to the control group. However, these effects were not as strong as those of rhPTH(1-34). Both PTH analogs demonstrated enhanced anabolic effects around the titanium implants, promoting bone regeneration and remodeling.”

Line 224 - The authors have attributed this phenomenon to the unique anatomical characteristics observed in the jawbone.- I would suggest writing it in the first person

We totally understood the reviewer’s comment. We have corrected the sentences as follows.

“The anabolic effects of both PTH analogs in this specific region may have been enhanced by the unique anatomical characteristics of the mandible, which we attribute to these improvements.”

Line 236 - The authors have attributed this phenomenon to the unique anatomical characteristics observed in the jawbone.- This is outdated as the label of two year limit of Forteo use was lifted by FDA in 2021

Thank you for your valuable comments regarding the FDA’s decision to lift the two-year limit on Forteo (teriparatide) use in 2021. We have revised sentences to reflect this recent information in FDA guidelines as follows.

“Despite the FDA's decision to remove the two-year treatment limit in 2021, which opens possibilities for broader clinical applications, there are still numerous challenges that need to be addressed. There are ongoing concerns about the potential long-term effects of extended use, including accelerated bone remodeling, possible hypercalcemic conditions, and heightened bone resorption”

Line 380-382 - bone volume (TV; mm3), trabecular number (Tb.N; 1/mm), trabecular thickness (Tb. Th; um), trabecular separation (Tb.sp; µm).- minor points- please superscript mm3, and change u -> µ

We appreciate reviewer’s detailed comments. We have corrected the part about unit display in figure legend.

Line 405-406 - following treatment with dimeric dimeric R25CPTH(1-34)

- please remove redundancy.

We removed dimeric duplication in the figure legend for figure 5 as follows.

“Figure 5. Measurement of biochemical Marker Dynamics in serum. The serum levels of calcium, phosphorus, P1NP, and CTX across three time points (T0, T1, T2) following treatment with dimeric R25CPTH(1-34), rhPTH(1-34) and control.”

Line 409-410 - CTX levels, associated with bone resorption, show no significant differences between groups.- there is a missing figure identification. please specify relevant figure - I guess (E)

We appreciate the reviewer's insightful comment regarding the missing figure identification in the sentence about CTX levels. After reviewing Figure 5, we have specified the relevant figure panel as follows:

“Figure 5. (A) The study timeline. (B-C) Calcium and phosphorus levels show an upward trend in response to both PTH treatments compared to control, indicating enhanced bone mineralization. (D) P1NP levels, indicative of bone formation, remain relatively stable across time and treatments. (E) CTX levels, associated with bone resorption, show no significant differences between groups.”